# Some Properties of Composite Drone Blades Made from Nanosilica Added Epoxidized Natural Rubber

**DOI:** 10.3390/polym12061293

**Published:** 2020-06-05

**Authors:** Sunisa Suchat, Aunnuda Lanna, Aujchariya Chotikhun, Salim Hiziroglu

**Affiliations:** 1Faculty of Science and Industrial Technology, Prince of Songkla University, Surat Thani Campus, Mueang, Surat Thani 84000, Thailand; sunisa.su@psu.ac.th (S.S.); Aunnuda_w.ch@hotmail.com (A.L.); aujchariya.c@psu.ac.th (A.C.); 2Department of Natural Resource Ecology & Management, Oklahoma State University, Stillwater, OK 74078-6013, USA

**Keywords:** drone blade propellers, epoxidized natural rubber, nanosilica, impact strength, accelerated weathering

## Abstract

The objective of this study was to investigate the basic properties of composite materials that were made from epoxidized natural rubber and nanosilica to be used as blades for drones. Nanocomposite samples were prepared with 5% of epoxidized natural rubber and epoxy resin loaded with 3% nanosilica. Their resistance against accelerated weathering conditions as well as mechanical properties, including flexural strength, impact strength, and hardness, were evaluated. Based on the findings of this work, the impact strength of the samples decreased 13.33% and 33.33% as a result of exposing them to weathering by UV radiation for 168 h and 336 h, respectively. However, their tensile strength properties enhanced 35.71% and 19.05% for the above corresponding exposure time spars. Experimental composite samples that were made in this study would have great potential to be used as raw material for propeller blade for drones based on their properties evaluated within the scope of this work.

## 1. Introduction

Currently, there has been a growing demand for Unmanned Aerial Vehicles (UAV) or drone to be used for different applications. Presently, the number of small hobbyist drones in the US was 1.1 million in 2016 and Federal Aviation Administration (FAA) estimates that such a number will reach more than 3.5 million by 2021. The agency also estimates the commercial drone fleet will grow from 42,000 to about 442,000 units by 2021 [1,2]. It is a fact that the global demand for aircraft is growing significantly and the aerospace industry is depending more and more on composite materials to reduce the weight as well as production cost [3,4]. The first generation of composites used in aircraft construction was in 1960s and 1970s [1,3]. Epoxy based resin systems that were converted in laminated structures had certain problems in terms of creating debris during service life in 1970s [5]. However, with great improvements, advanced epoxy resins systems had desirable characteristics to be used in aircraft industry later decades [4,6].

Demand and use of composites as well as epoxy based resins in the manufacture of drones are also part of an increasing trend. The aviation safety agency stated that there could be as many as 1.6 million commercial drones in use by 2021. UAV were very popular due to their ability to access areas quickly and accurately. The UAV mainly originated in military applications, but its use has rapidly expanded to commercial, scientific, agricultural, and other fields. The overall design and material selection in drones are main players in their effective and efficient use. The employment of different types of composite products is widely used in drone production. Such composites, including polymer based origin, are also used for the propeller manufacture of a drone. When a typical UAV is in flight, debris can impact the fast-moving propeller blades and disable the UAV, thus creating problems. It is a well-known fact that polymer based composite materials have advantages over other materials regarding their strength characteristics. Therefore, they could have great potential to be used in production of propeller blades of the drones. In general, some UAV components, primarily propeller, are manufactured using polymeric composite materials, such as carbon fiber, nylon, fiber glass, and stainless titanium, due to their excellent properties, including high strength and light weight. However, epoxy based resins have unique characteristics among these composites and they are used as the adhesive matrix with the fibers to have enhanced strength and smooth glossy surfaces [2,3,4,5]. Epoxy resins are currently used extensively because of their superior properties, such as high modulus, low creep, and reasonable elevated temperature performance. However, they easily fail under impact because of their highly cross-linked structure [6,7,8]. Certain chemicals and materials have been added into epoxy including hard particulate materials, such as inorganic glass particles and nanoparticles, in order to improve its toughness [9,10,11,12,13].

During the last decade, nanomaterials, including nanosilica and carbon nanotubes, are widely used for many applications. The main objective of adding nanoparticles in a member is to enhance its overall properties. Nanosilica has also been used in various applications, ranging from the manufacture of experimental samples of cement board to different types of wood composite panels with enhanced structural properties in past studies [11].

Natural rubber is produced from rubber tree (*Hevea brasiliensis*), which is one of the most popular plantation species in South East Asian countries, including Thailand and Malaysia. These two countries are the main producers of natural rubber in South East Asia. Natural rubber (NR) contains epoxy groups that can be modified into epoxidized natural rubber (ENR) at several degrees, namely 25%, 50%, and 75%, referred to as ENR-25, ENR-50, and ENR-75, respectively [12].

In a previous study, different blends of elastomers and thermoplastics were studied from the perspective of influence of epoxidized natural rubber (ENR) on properties of such mixtures as a function of rubber content [13]. It was suggested that the impact strength of epoxy resin can be improved by blending with ENR-50, resulting in higher impact strength than that of ENR-25 [13,14]. It was also determined that the use of such raw material would have potential in manufacture of Unmanned Aerial Vehicles (UAV) [14]. Therefore, it appears that ENR-50 would be considered to be an ideal choice for improving the impact resistance of epoxy resin for use different elements of UAV, specifically for propeller blades [14,15]. With the ever-increasing use of advanced composite materials, the influence of chemical or environmental aging is caused by various agents, such as humidity, wind, and ultraviolet radiation (UV) cause to irreversible changes in the molecular structure of units [16]. Consequently their overall durability is adversely influenced, which has become a major current concern to the aeronautic industry [17,18,19]. Some of the previous studies suggest that tensile and compressive properties of carbon-epoxy composites may improve during aging at the initial consolidation stage. In this case, the improvement in mechanical properties of the member was attributed to post-curing reactions [20]. A degradation stage followed when the composite properties are significantly decreased, which could be attributed to a deterioration of the matrix and weakening of the fiber-matrix interface [20,21,22]. Although various studies regarding the aging of epoxy composites are available in the literature [17,18,19,20], the effects of aging and the extent of those parameters on properties of these materials are not fully understood and there is very limited information in this area [23,24]. Therefore, the objective of this study was to evaluate the weathering resistance of nanocomposites for UAV applications made of epoxy resin with ENR and nanosilica exposed to alternating cycles of UV-A radiation and water condensation in an accelerated aging chamber. Additionally, the influence of aging on the material properties was evaluated by employing Fourier-Transform Infrared (FTIR) Spectroscopy. Additionally, mechanical properties, namely the tensile strength, impact strength, and hardness of the samples, were also tested. Finally, thermogravimetric (TGA), and Scanning Electron Microscopy (SEM) analysis of the experimental samples were carried out in this work to have a better understanding of the basic behaviors of such value-added composites.

## 2. Materials and Methods

Epoxidized natural rubber (ENR-50) was obtained from Muang Mai Guthrie Co. Ltd., Bangkok, Thailand. The Bisphenol-type epoxy resin (Epicholrohydrin–bisphenol A: Epotec YD 535 LV, number average molecular weight <700) was purchased from Aditya Birla Chemical Industry Co., Ltd., Bangkok, Thailand. The polyamide resin (TH 7255) that was used as curing agent was supplied by Siam Chemical Industry Co., Ltd. Bangkok, Thailand. The nanosilica colloidal in the form of white nanosilica powder surface-modified with methyl triethoxy silane was supplied by Bossofticl Co., Ltd., Bangkok, Thailand. Nanosilica colloidal with chemical formula of SiO_2_ typically has a particle size within the range of 1 to 5 nm. The average molecular weight of the nanosilica was 60 g/mol, with a specific surface area of 135 m^2^/g. The 40 wt% nanosilica colloidal was lightly suspensded in H_2_O [22,23,24].

The epoxy nanocomposite was loaded with 5% ENR-50 and 3% nanosilica by mechanical stirrer. ENR-50 were stirred at 5 phr (parts per hundred parts of epoxy resin by weight) before nanosilica and epoxy resin were stirred for 15 min. at 50 rpm at a temperature of 25 °C. Next 15 min., the curing agent (35 phr) was added and the mixture was stirred for 5 min. A total of 135 samples, three replications of each rubber molds having 45 samples for each test were used for the experiments.

### 2.1. Properties of the Samples

Fourier-Transform Infrared (FTIR) Spectroscopy of ILSS samples was performed using a Shimadzu IRTracer-100 Spectrometer to compare the spectra of raw material and nanocomposite. The analysis conditions were 32 scans, at a range of 400–4000 cm^−1^ resolution, being analyzed in Attenuated Total Reflectance (ATR) mode

### 2.2. Accelerated Aging Test of the Samples

The accelerated aging of the samples was performed following ASTM G154 cycle I while using a QUV-accelerated weathering tester (Q-UV test LU-0819, Q-panel lab products, USA). The condition of QUV-accelerated weathering aging testing includes UVA radiation wave length of 340 nm and intensity of 0.77 W/m^2^ at 60 °C for 8 h and water-condensation at 50 °C for 4 h and spray water on the specimen for 15 min. The aging times were varied at 168 h and 336 h.

The testing of the samples was performed following ASTM G154 cycle VII using a QUV-accelerated weathering tester (Q-UV test LU-0819, Q-panel lab products, USA). The condition of QUV-accelerated weathering aging testing includes UVB radiation wave length of 340 nm and intensity of 1.55 W/m^2^ at 60 °C for 8 h, spray water on the specimen for 15 min, and water-condensation at 50 °C for 4 h. The aging times were varied at 168 and 336 h.

### 2.3. Mechanical Properties of the Samples

The izod impact test was conducted with a cantilever impact tester at room temperature, according to ASTM D256. The specimen dimensions were 63.5 × 12.7 × 3.0 mm. The tensile properties of the samples were carried out on a Lloyd LR 10K Universal Testing machine that was equipped with a 10 kN load cell, using crosshead speed of 50 mm/min. according to ASTM D638. The hardness of the samples with a minimum 6.0 mm thickness was also determined according to ASTM D2240. The Izod impact strength of the cured nanocomposites specimen had a V-shaped notch and samples were measured according to ASTM D256. The specimen dimensions were 63.5 mm × 12.7 mm × 3.0 mm and the depth under the notch of the specimen was 10.2 mm. The tests were carried out at room temperature and the values were taken from an average of five specimens.

### 2.4. Flexural Strength and Thermal Properties of the Samples

ASTM D790 was used to determine the bending characteristics of the samples with a recommended span to depth ratio of 16:1. The specimen had 127 × 12.5 × 3 mm (length × width × thickness). The flexural test was conducted on the same equipment with crosshead speed rate of 1.2 mm/min.

Thermogravimetric analysis (TGA) of the samples was also carried out at temperature levels ranging from 30 °C to 900 °C in N_2_ and O_2_ air, with a heating rate of 10 °C min^−1^ and flow rate of 100 mL/min while using about 10 mg for each sample.

## 3. Results and Discussion

### 3.1. FTIR Analysis of the Samples

Fourier transform infrared spectroscopy (FTIR) was employed in order to determine the epoxide group of the samples. The amount of epoxide groups in natural rubber in situ epoxidation was evaluated by Fourier transform infrared spectroscopy. Figure 1 depicts peak values at the wavenumber of 873 cm^−1^ (C-O-C, asymmetric stretching). The characteristic peak at 873 cm^−1^ stands for the asymmetric stretching vibration of oxirane ring [17]. The FTIR results of the produced colloidal nanosilica displayed in the line 2 the peaks appearing in the range of 450 cm^−1^ to 1300 cm^−1^ are typical of silica samples. The FTIR results confirmed that the three main characteristic peaks of silica at 1093 cm^−1^, 788 cm^−1^, and 466 cm^−1^ attributed to the asymmetric, symmetric stretching vibration, and the bending modes of silica, respectively [18].

Moreover, line 3 in Figure 1 illustrates the nanocomposite from epoxy resin, epoxidized natural rubber, and nanosilica. Chemical structures of epoxy resin and the peaks characteristic bands that belong to C-H vibrations are observed in the regions of 2830–3000, 1600, 1400–1450, and 1375 cm^−1^ for aliphatic C-H stretching aromatic C-C stretching, aliphatic C-H bending, and C-H rocking, respectively. Additionally, the presence of epoxide groups is characterized by the appearance of typical bands at 1250, 913, and 830 cm^−1^, corresponding to stretching C-O-C of ethers, stretching C-O of oxirane group, and stretching C-O-C of oxirane group [19]. Additionally, the peaks appearing in the range of 450 cm^−1^ to 1300 cm^−1^ are typical of silica samples [18]. The FTIR results confirmed the three main characteristic peaks of silica at 1093 cm^−1^, 788 cm^−1^, and 466 cm^−1^ attributed to the asymmetric, symmetric stretching vibration, and the bending modes of silica, respectively.

### 3.2. Mechanical Properties of Nanocomposite Samples

Table 1 lists tensile strength values of the samples. The tensile strength of epoxy nanocomposites revealed that accelerated aging improved tensile strength of epoxy nanocomposites as shown in Figure 2. This could be due to epoxy resin that can be cured by UV cure. The specimens absorb the UV light from QUV-accelerated weathering tester caused increased crosslink resulting in an enhancement of their hardness. However, the addition of ENR and nanosilica in epoxy resin showed a higher tensile strength than that of pure epoxy resin. Such a finding would be due to the small particle size of ENR and nanosilica having a good compatibility and dispersion in the epoxy matrix, which can be attributed to reinforcing filling ability of nanosilica.

The impact strength values of epoxy nanocomposite samples after the Izod impact test in each condition were evaluated and are displayed in Table 2. After accelerated aging exposure, the surfaces of samples exhibited a distinct change in color from colorless to dark yellow. The results of an increase time in the accelerated aging chamber for 168 h and 336 h, which reduced the impact strength of samples, are shown in Figure 3. However, using UVA radiation gave a higher impact strength than UVB radiation, because UVA radiation gave a wave length shorter than UVB radiation.

The TGA thermograms of samples in all of the conditions were evaluated, as depicted in Figure 4. Residual weight was between 6–7% and the maximum degradation temperature (T_m_) was around 370 °C for all conditions, the maximum degradation temperature was obtained using derivative thermogravimetry (DTG). However, using UVB radiation gave higher thermal stability than UVA radiation which could be due to the specimen get light UV from QUV-accelerated weathering tester caused increased crosslinking. At the final 600 °C temperature, the fillers nanosilica in the composites might be partly decomposed in the second step and some of the residue would still remain, and its amount tended to increase with filler. It can be noted that the TGA was performed at temperature levels that ranged from 30 to 600 °C, ranging from 600 to 900 °C under nitrogen and oxygen atmospheric conditions, respectively. Two decomposition steps can clearly be observed in Figure 4. The first step is for degradation under nitrogen atmosphere and the second step is for decomposition under oxygen atmosphere.

It appears that samples had the optimum formulation of epoxy resin 100 phe, curing agent 35 phe, ENR-50 phe, and nanosilica 3 phe in order to evaluate their mechanical properties.

### 3.3. Impact Properties of Nanocomposite-Fiber Samples

Nanocomposite samples that are molded together with nanofiber were tested for weathering resistance by the accelerator chamber for 168 h while using UV-B light. After accelerated aging, improved impact of nanocomposites-fiber increased relatively, as can be seen in Figure 5. This could be due to epoxy resin that can be cured by UV. The specimens absorbed UV light from QUV-accelerated weathering tester increasing crosslink, so that the overall hardness of the samples improved. When considering the ability of energy absorption of the specimen, it was found that the addition of nanosilica showed the highest energy absorption capacity as compared with epoxy. This could be due to the crystallization of nanosilica creating holes or gaps, which can help absorption of higher magnitude of energy by the sample. Absorbed energy in breaking point of the specimen can be expressed in SI units and it was determined by the impact tester machine. The Izod pendulum impact test records amount of the energy to break composite specimens of specified size based on ASTM D256. The speed of the impact depends on the length of the arm, and the height from which it is dropped. The difference in height between the starting position and the end position is used to calculate the difference in energy, which is the energy that was absorbed by the test specimen as it broke.

### 3.4. Flexural Strength of Nanocomposite-Fiber Samples

Figure 6 illustrates the flexural strength of the samples. The average value of flexural strength of weathered and unweathered epoxy samples had 285 MPa, while the corresponding value was 337 MPa for those nanosilica added samples. It seems that the addition of nanosilica into the specimens enhanced their overall flexural strength characteristics, which could be related to good dispersion compatibility and fillers.

### 3.5. Hardness of Nanocomposite-Fiber Samples

Figure 7 shows the hardness of the nanocomposite samples after testing for resistance. It was found that, after testing the resistance to nanomaterials, the value increased when compared with the specimens stored in normal conditions due to UV-B light causing the nanomaterials to deteriorate. This resulted in increased sample hardness and, when comparing the hardness values obtained, were not significantly different.

### 3.6. Microscopic Evaluation of Nanocomposite-Fiber Samples

The SEM micrographs of notched Izod impact fracture surfaces of composite materials were assessed, as can be seen in Figure 8. It has been generally recognized that one of the major factors affecting the mechanical properties of nanofiber based composites is the fiber/matrix interphase. Stress transfer in the fiber/matrix interphase requires a strong interfacial bond between the two components [20]. Figure 8a presents numerous cavities existed after the fibers were pulled out of the matrix. The presence of these holes can be related to weak interfacial bonding between the fiber and the matrix polymer resulting in inefficient stress transfer from the matrix. The fiber pulled out results in the generation of flaws, which creates voids between the fibers and the epoxy matrix. Flexural and impact strength of the epoxy based samples were determined to be lower than those of having nanoparticles, as illustrated in both Figure 5 and Figure 6. It appears that the adversely influenced mechanical properties of the samples could be due to weak interfacing stated above.

## 4. Conclusions

Based on the finding in this work, it appears that epoxidized natural rubber and nanosilica improved the impact strength of epoxy nanocomposite as reinforced materials. Even though the impact strength values were reduced by accelerated aging after 168 h and 336 h, those values were generally higher than epoxy resin composites. The tensile strength values of the specimens were increased when using a QUV-accelerated weathering both UVA and UVB radiation. Therefore, the nanocomposites of those experiments illustrated a good weathering resistance for outdoors UAV blade applications. In further investigation, it would be desirable to evaluate the mechanical and physical properties of the samples manufactured by the addition of different types of fiber, such as carbon fiber, to have a better understanding efficient and effective use of such units in drone manufacture.

## Figures and Tables

**Figure 1 polymers-12-01293-f001:**
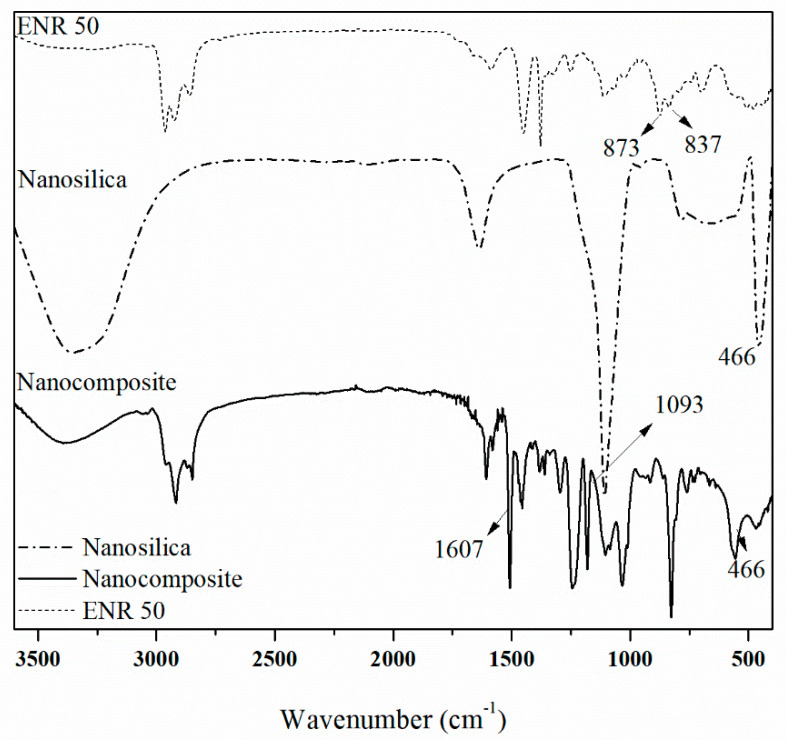
Fourier-Transform Infrared (FTIR) spectra of nanocomposite samples.

**Figure 2 polymers-12-01293-f002:**
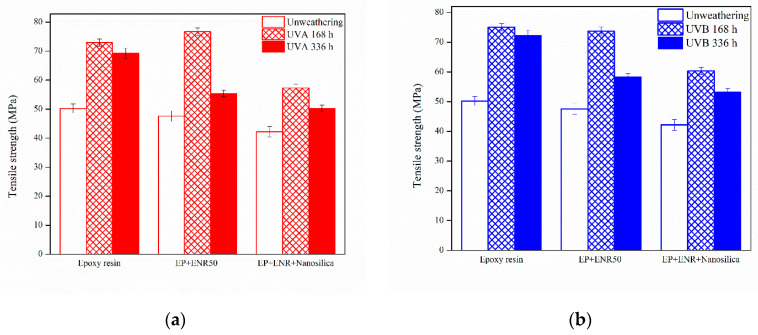
Tensile strength of nanocomposites after accelerated aging (**a**) UVA and (**b**) UVB.

**Figure 3 polymers-12-01293-f003:**
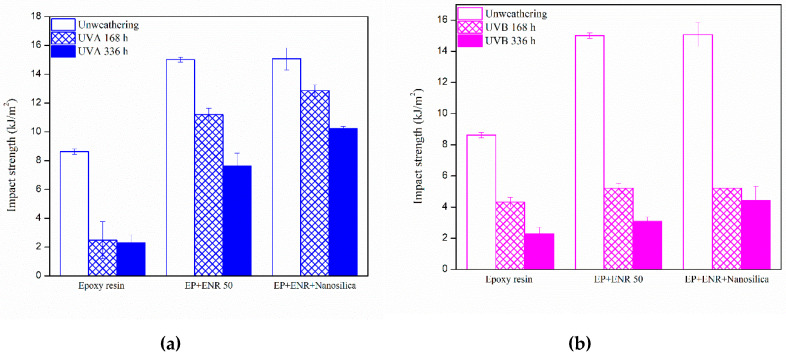
Impact strength of nanocomposites after accelerated aging (**a**) UVA and (**b**) UVB.

**Figure 4 polymers-12-01293-f004:**
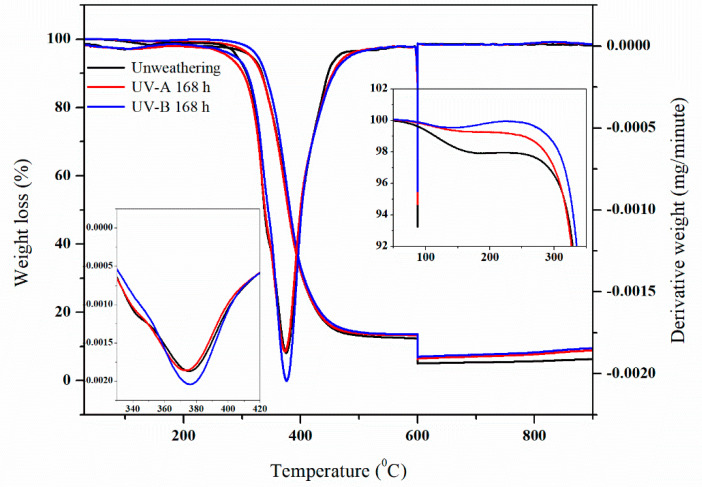
Thermogravimetric analysis (TGA) Thermogram of nanocomposites.

**Figure 5 polymers-12-01293-f005:**
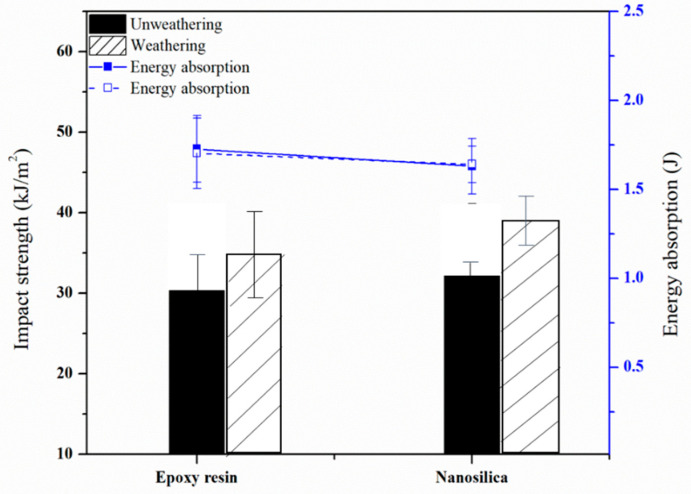
Impact strength of nanocomposites-fiber samples after accelerated aging.

**Figure 6 polymers-12-01293-f006:**
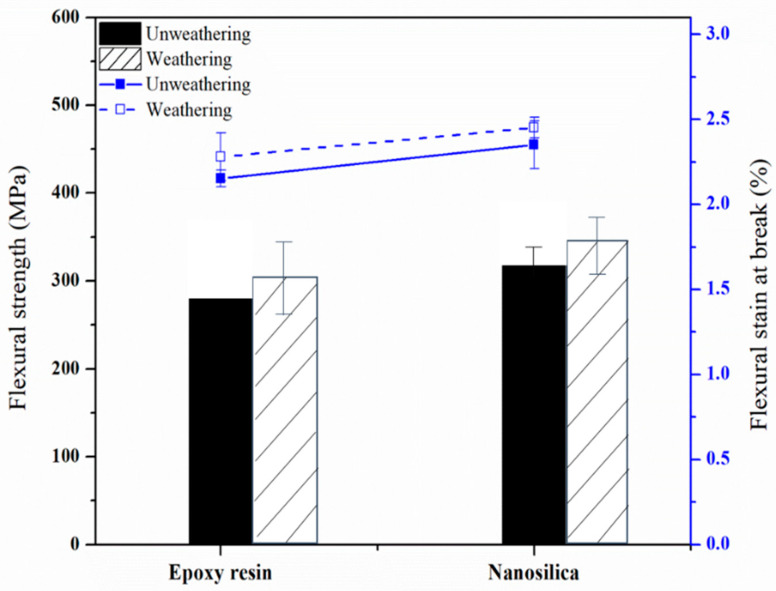
Flexural strength of nanocomposites-fiber samples after accelerated aging.

**Figure 7 polymers-12-01293-f007:**
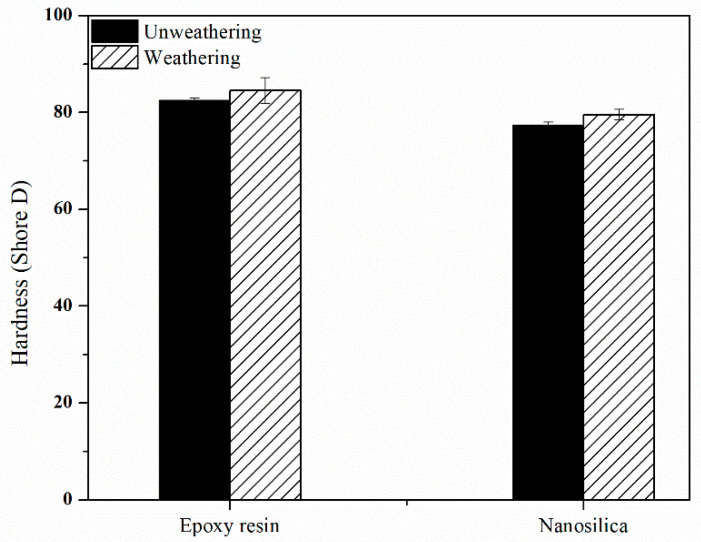
Hardness of nanocomposite-fiber samples after accelerated aging.

**Figure 8 polymers-12-01293-f008:**
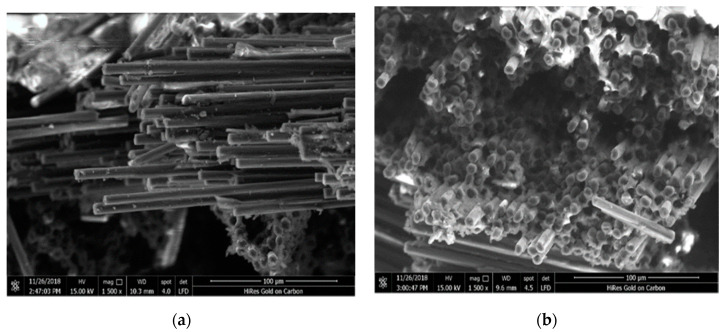
Scanning Electron Microscopy (SEM) micrographs of fracture surfaces of (**a**) composite-fiber epoxy resin sample and (**b**) nanocomposite-fiber sample.

**Table 1 polymers-12-01293-t001:** The mean values of the tensile strength of the samples.

Formula	Tensile Strength (MPa)				
	Non-Weathering	168 h-Weathering		336 h-Weathering	
		UVA	UVB	UVA	UVB
Epoxy resin	50.22 (0.56)	72.95 (0.21)	73.50 (0.11)	69.34 (0.72)	70.34 (2.45)
EP+ENR-50	47.61 (0.82)	76.71 (0.33)	77.21 (0.29)	55.37 (0.13)	57.79 (0.34)
EP+ENR+Nanosilica	42.19 (0.83)	57.32 (0.24)	57.50 (0.17)	50.21 (0.15)	50.48 (0.65)

The numbers in parentheses are the standard deviation values.

**Table 2 polymers-12-01293-t002:** The mean values of the impact strength of the samples.

Formula	Impact Strength (kJ/m^2^)				
	Non-Weathering	168 h-Weathering		336 h-Weathering	
		UVA	UVB	UVA	UVB
Epoxy resin	8.62 (0.18)	2.48 (1.29)	4.42 (0.30)	2.31 (0.55)	2.28 (0.42)
EP+ENR-50	15.06 (0.18)	11.21 (0.43)	5.21 (0.33)	7.64 (0.87)	3.09 (0.28)
EP+ENR+Nanosilica	15.07 (0.76)	12.86 (0.40)	5.21 (0.57)	10.23 (0.13)	4.43 (0.91)

The numbers in parentheses are the standard deviation values.

## Data Availability

The data used to support the findings of this study are available from the corresponding author upon request.

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
