# Peer review of "Some Properties of Composite Drone Blades Made from Nanosilica Added Epoxidized Natural Rubber"

_polymers, 2020, doi:10.3390/polym12061293_

Round 1

Reviewer 1 Report

The authors have written an interesting paper that reports work on evaluating the weathering resistance of nanocomposites for UAV applications made of epoxy resin with ENR and nanosilica exposed to alternating cycles of UV-A radiation and water condensation in an accelerated aging chamber. 

  1. English should be improved in some parts of the paper.
  2. The abbreviation “NR” at line 119 should be clarified.
  3. Please explain the ENR 50, ENR 25 in the introduction part.
  4. References are missing for the phrase “With the ever-increasing use of …. units” from line 54 to line 57.
  5. The phrase “there is very listed information in this area” at line 65 should be recheck. Maybe it should be “there is very least information in this area”.
  6. Please add the producer and type of nanosilica. In addition, chemical composition and physical properties of materials should be clarified.
  7. From line 85 to 86, the sentence “Properties of the samples…” are still uncompleted.
  8. Please clarify the quantity of samples use in the present work. In addition, the sample for Izod test come with notch or not, it should be described.
  9. Please explain the peak around 600 oC in TGA result.
  10. 5 should be made clearer, the energy absorption graph attributing to the exposing condition should be clarified.
  11. Please discuss the relationship between the found in microstructure and the change in mechanical properties of the present materials.
  12. Please check the conclusion at line 221 to 222, “Furthermore, this composites can be mixed with co-carbon fiber to increase flexural strength of drone blades which is informal to broken in case of crash.” Because there is no data concern to the corporation of the present materials with carbon fibers.

Reviewer 2 Report

I don't think there exist sufficient innovation for this manuscript to be published on Journal of Polymers, at least I didn't see any attractive description in Introduction part.

The data is not enough for publication. Authors should prepare more samples to evaluate the composites.

Round 2

Reviewer 2 Report

I didn't see the responds of my comments.

The responds I saw in the manuscript was from other reviewer.